# Design and Analysis of Lithium–Niobate-Based Laterally Excited Bulk Acoustic Wave Resonator with Pentagon Spiral Electrodes

**DOI:** 10.3390/mi14030552

**Published:** 2023-02-26

**Authors:** Ying Xie, Wenjuan Liu, Yao Cai, Zhiwei Wen, Tiancheng Luo, Yan Liu, Chengliang Sun

**Affiliations:** 1The Institute of Technological Sciences, Wuhan University, Wuhan 430072, China; 2Hubei Yangtze Memory Laboratories, Wuhan 430205, China

**Keywords:** XBAR, LiNbO_3_ thin film, effective electromechanical coupling coefficient

## Abstract

In this paper, we present a comprehensive study on the propagation and dispersion characteristics of A_1_ mode propagating in Z-cut LiNbO_3_ membrane. The A_1_ mode resonators with pentagon spiral electrodes utilizing Z-cut lithium niobate (LiNbO_3_) thin film are designed and fabricated. The proposed structure excites the A_1_ mode waves in both *x*- and *y*-direction by utilizing both the piezoelectric constants *e*_24_ and *e*_15_ due to applying voltage along both the *x*- and *y*-direction by arranging pentagon spiral electrode. The fabricated resonator operates at 5.43 GHz with no spurious mode and effective electromechanical coupling coefficient ($K_{eff}^{2}$) of 21.3%, when the width of electrode is 1 µm and the pitch is 5 µm. Moreover, we present a comprehensive study of the effect of different structure parameters on resonance frequency and $K_{eff}^{2}$ of XBAR. The $K_{eff}^{2}$ keeps a constant with varied thickness of LiNbO_3_ thin film and different electrode rotation angles, while it declines with the increase of *p* from 5 to 20 µm. The proposed XBAR with pentagon spiral electrodes realize high frequency response with no spurious mode and tunable $K_{eff}^{2}$, which shows promising prospects to satisfy the needs of various 5 G high-band application.

## 1. Introduction

In recent years, with the explosion of mobile data from video streaming, virtual reality, and wireless communication, the needs for high frequency and large bandwidth of radio-frequency (RF) components have increased dramatically [1,2,3]. Nowadays, the surface acoustic wave (SAW) resonator and the thin film bulk acoustic wave resonators (FBARs) have dominated the market due to the excellent performances. However, there are obstacles for them to be used to high-frequency and large-bandwidth RF front-end devices. On the one hand, the effective electromechanical coupling coefficient ($K_{eff}^{2}$) of both SAW and FBAR are limited to 6–13% with aluminium nitride (AlN) thin film, not satisfying the demand of the relative bandwidth of filter above 6%, such as band N77 and N79. On the other hand, the SAW resonators and FBARs are difficult to operate above 5 GHz. The frequency of SAW hardly amounts to 3.5 GHz due to the limitation of acoustic velocity and the lithography technology. For FBAR, the thinner the piezoelectric film, the higher the frequency. However, the thinner thickness of piezoelectric film may cause the degradation of film crystal quality, which eventually influences the performance of devices [4,5,6].

In recent years, the laterally excited bulk acoustic resonator (XBAR) devices based on lithium niobate (LiNbO_3_) thin film have been extensively studied as promising candidates for the application in fifth generation mobile communication [7,8,9,10,11]. XBARs can achieve a high frequency of more than 5 GHz, and a large $K_{eff}^{2}$ of more than 20%, exceeding the traditional FBARs and SAW resonators due to the larger piezoelectric coefficients *e_15_* and *e_24_* of LiNbO_3_. In 2019, V. Plessky et al. first presented a 4.8 GHz XBAR based on Z-cut LiNbO_3_ thin film with a $K_{eff}^{2}$ of 25%, showing the possibility of its application to filters in the 3–6 GHz range [12]. In 2020, Ruochen Lu presented XBAR in 128° Y-cut LiNbO_3_ thin films with the $K_{eff}^{2}$ of 46.4%, which obtained the highest value of $K_{eff}^{2}$ [13]. In 2021, Bohua Peng designed and fabricated a solid-mounted-type XBAR on ZY- LiNbO_3_, operating at 5 GHz, to improve heat dissipation and temperature coefficient of frequency (TCF) [14].

The $K_{eff}^{2}$ of resonator has a significant influence on the bandwidth of filters, and it can be adjusted by structural optimization and tuning piezoelectric coefficients. For example, Gianluca Piazza investigated the influence of the electrical boundary conditions of Lamb wave resonator imposed by the excitation electrodes on the $K_{eff}^{2}$, and determined that $K_{eff}^{2}$ can be tuned with a varying range from 3% to 7% [15]. Jie Zou investigated the impact of Euler angle of LiNbO_3_ film on the $K_{eff}^{2}$ of the resonator. The $K_{eff}^{2}$ varies largely due to the prominent anisotropy of the piezoelectric matrix, so the optimal cut angle can be chosen so as to optimize the $K_{eff}^{2}$ [16]. Compared with Lamb wave resonator, the research on the structure and vibrate modes of XBAR is relatively immature, and there is almost no systematical analysis of the effect of XBAR structure parameters on the $K_{eff}^{2}$.

In this paper, we present a comprehensive study on the propagation characteristics of the plate modes propagating in Z-cut LiNbO_3_ membrane. Then, we propose new pentagon spiral electrode structure to excite the A_1_ mode in the LiNbO_3_ thin film. The dependance of $K_{eff}^{2}$ with the electrode rotation angle and the pitch of electrodes are investigated. A series of XBARs with pentagon spiral electrodes with different electrode pitches are fabricated. The resonance frequency of fabricated device is around 5.4 GHz, and the $K_{eff}^{2}$ of the fabricated devices varied with the electrode pitch, corresponding to the results of simulations. The resonator with pentagon spiral electrodes operates at 5.4 GHz with tunable $K_{eff}^{2}$, showing promising prospect for application in super high-frequency RF front-end filters.

## 2. Design and Simulation

A.Propagation characteristics of plate mode

The plate waves propagating in the LiNbO_3_ thin film include either the symmetric and antisymmetric Lamb wave (S_0_, S_1_, A_0_ and A_1_) or the plate shear wave (SH_0_ and SH_1_). XBAR consists of a suspended LiNbO_3_ thin film with interdigital electrodes (IDEs) on top. When applying a voltage, with the generated lateral electric field in the electrode arrangement direction, shear antisymmetric A_1_ mode is generated. To more accurately capture the A_1_ mode propagation characteristics in Z-cut LiNbO_3_ thin film, the three-dimensional eigenfrequency simulation is set up to calculate the open phase velocity ($v_{p}$) and the coupling coefficient ($k^{2}$) dispersion of the plate waves propagating in the piezoelectric membrane [17,18,19]. The $v_{p}$ characteristics are the open-surface phase velocity and are calculated by Formula (1), and $k^{2}$ characteristics are calculated by Formula (2):(1)$$v_{p}=f\times \lambda ,$$
(2)$$k^{2}=\frac{v_{p}^{2}-v_{m}^{2}}{v_{p}^{2}},$$
where f is the eigenfrequency of the acoustic mode obtained by the FEM simulation, λ is the length of the acoustic mode, and the *v_m_* is the short-surface phase velocity calculated by FEM approach. In the simulation, the model is a 3D building block of the LiNbO_3_ plate with period boundary conditions on both *x*- and *y*-directions. The IDTs are assumed to be infinitely thin with no material assigned, which means the mechanical loading is ignored and only electrical boundary conditions considered herein.

Figure 1 depicts the mode shapes of the six plate modes. The displacement profile or the mechanical vibration of the Lamb wave modes is in the propagation *xz*-plane, and for the SH plate modes in the sagittal *yz*-plane. The $v_{p}$ and $k^{2}$ dispersion characteristics of six plate waves propagating in the Z-cut LiNbO_3_ membrane are shown in Figure 2, where the $h_{\mathrm{LiNbO}3}$ is the thickness of LiNbO_3_ membrane and λ represents the length of acoustic wave. Obviously, higher plate wave modes exhibit large $v_{p}$ with strong dispersion especially when the $h_{\mathrm{LiNbO}3}/\lambda$ is small, and high $v_{p}$ is generally favored for high frequency applications. As shown in Figure 2b, the frequencies of higher plate modes stop scaling with *β*, rather depending solely on the plate thickness. This indicates that for the A_1_ mode at low $h_{\mathrm{LiNbO}3}/\lambda$, the frequency is not pitch-controlled any more but thickness-controlled, which breaks the limit of lithography technology to frequency. High $v_{p}$ beyond 40,000 m/s and large $k^{2}$ of higher than 30% can be obtained as $h_{\mathrm{LiNbO}3}/\lambda$
*<* 0.1, making the feasibility of wideband operating in 6G bands using A_1_ mode resonators in Z-cut LiNbO_3_.

B.Resonator design

The proposed XBAR with pentagon spiral shape electrode is illustrated in Figure 3a. The pentagon spiral electrodes are arranged alternatively on the top of Z-cut LiNbO_3_ thin film, with two thin anchors connecting to the Ground–Signal–Ground (GSG) pads. The electrical potentials are alternatingly applied to adjacent electrodes, as illustrated by “+” and “−” signs indicated in Figure 3, creating electric fields along both the *x*-direction and *y*-direction. The A-A’ cross-section view of the resonator is shown in Figure 3b, and an air cavity is formed underneath the LiNbO_3_ thin film via backside release silicon substrate and silicon dioxide (SiO_2_).

The finite element simulation is carried out to further analyze the performance of the resonator with pentagon spiral electrodes. The thickness of the Z-cut LiNbO_3_ thin film (*h_p_*) and electrode (*h_e_*) is set to 330 nm and 200 nm, respectively, to achieve a 5.4 GHz resonant frequency (*f_s_*), as shown in Figure 4a. The width of electrodes (*w*) is defined as 1 µm and the pitch (*p*) equals to 5 µm. Figure 4b shows the horizontal displacement distribution at resonance frequency along the A-A’ cross-section of the resonator with pentagon spiral electrodes and shows the horizontal displacement deformation in the B-B’ cross-section. The horizontal displacement is antisymmetric with the centre plane of the wave excited in the piezoelectric layer, which corresponds to the standard A_1_ mode acoustic wave [20,21]. The proposed pentagon spiral-shaped electrodes applying alternating voltage can excite the electrical fields along both the *x*-direction and *y*-direction, thus exciting A_1_ mode shear wave along both *x*- and *y*-directions, utilizing both the *e*_24_ and *e*_15_ piezoelectric constants [19]. However, the $K_{eff}^{2}$ of structure with pentagon spiral-shaped electrodes is not improved relative to that of IDTs structure, which may be owing to the piezoelectric coefficient being *e*_24_ equal to *e*_15_, and no superposition effect occurs. Furthermore, we investigate the influence of the electrode rotation angle in respect to *x*-direction and the influence of *p* on the $K_{eff}^{2}$.

First, we investigate the effect of electrode rotation angle in respect to *x*-direction with Z-cut LiNbO_3_ plate on $K_{eff}^{2}$. The $K_{eff}^{2}$ of XBAR can be obtained by the approximated Formulas (2) and (3) [22,23,24]. It can be concluded that the $K_{eff}^{2}$ is positive to piezoelectric coefficient *e*_15_ of Z-cut LiNbO_3_ thin film. The simulated $K_{eff}^{2}$ vs. different rotation angle is depicted in Figure 5. The $K_{eff}^{2}$ of the XBAR is maintained around 23.5% as the rotation angle of electrodes increases from 0° to 180°, which may contribute to the equality of the *e*_24_ and *e*_15_ of Z-cut LiNbO_3_.
(3)$$K_{eff}^{2}=\frac{K^{2}}{1+K^{2}}=\frac{\pi^{2}}{4}\times \frac{\left(f_{p}-f_{s}\right)}{f_{p}},$$
(4)$$K^{2}=\frac{e_{15}^{2}}{\varepsilon_{r}\varepsilon_{0}C_{44}},$$
where the $f_{s}$ and $f_{p}$ are the resonance and anti-resonance frequency of the resonator, *e*_15_ is the piezoelectric coefficient of the LiNbO_3_ thin film, $\varepsilon_{r}$ and $\varepsilon_{0}$ are the relative permittivity and vacuum permittivity, and $C_{44}$ is the elastic constant of the LiNbO_3_ thin film.

By simplifying the classic dispersion equation for anti-symmetric A_1_ mode and using $h_{p}/p$ as a small parameter, we can obtain the approximate dispersion Equation (5) [25]. More precisely, for practical design of XBARs, it is advised to use the empirical Formula (6) [26].
(5)$$\frac{\Delta \omega }{\omega_{s}}={\left(\frac{h_{p}}{p}\right)}^{2}\times \left\{\frac{1}{2}+\frac{8}{\pi }\times \frac{V_{T}}{V_{L}}\times \frac{1}{\tan\left(\pi \frac{h_{p}}{\lambda_{L}}\right)}\right\},$$
(6)$$\frac{\Delta \omega }{\omega_{s}}=C1\times \frac{h_{p}}{p}+C2\times {\left(\frac{h_{p}}{p}\right)}^{2}.$$

Here, $\omega_{s}=2\pi f_{s}$ is the angular frequency of resonator. The first term “1/2” in brackets corresponds to simple “shear wave in rectangular resonator” model, while the second term shows that longitudinal component of Lamb wave does matter. The $\lambda_{L}$ is the wavelength of longitudinal bulk wave dependent on frequency. C1 and C2 are constants for linear and quadratic coefficients. According to the empirical Formula (4), we describe the relationship between $\Delta \omega /\omega_{s}$ and *h_p_*/*p*. It can be deduced from Formula (5) and Figure 6 that $\Delta \omega /\omega_{s}$ is negatively correlated with *p*, which indicates that the decrease of $K_{eff}^{2}$ with the increase of *p* as $h_{p}$ is the same.

## 3. Fabrication and Results

A series of XBARs, utilizing pentagon spiral electrodes with varied *p* and utilizing IDEs, have been fabricated. The fabrication process of resonator is shown in Figure 6a. The substrate wafer consisted of 330 nm-thick Z-cut LiNbO_3_ thin film, 2 µm-thick SiO_2_ and Si substrate, which is provided by NanoLN. Inc. First, 200 nm Mo thin film is deposited on the surface of LiNbO_3_ thin film and patterned as pentagon spiral shape or IDEs by lithography and reactive ion etching technology. Then, 300 nm-thick SiO_2_ is deposited on the surface of LiNbO_3_ and Mo thin film by Plasma Enhanced Chemical Vapor Deposition (PECVD) as protective layer. Subsequently, LiNbO_3_ thin film release is performed with backside Si deep reactive-ion etch (DRIE) process, followed by a wet etch with hydrofluoric acid solution to remove the buried SiO_2_ layer underneath the piezoelectric membrane. By exactly controlling the release time, the resonators with suspended working area only are realized. The scanning electron microscope (SEM) images of the fabricated devices with pentagon spiral shape and IDEs are shown in Figure 6b,c.

The scattering (S) parameter measurements are carried out using Keysight Network Analyzer (N5222B) connecting to a Cascade Microtech’s GSG probe station. Prior to the measurement, the setup is properly calibrated to remove the contribution of the probes and the cabling and only measure the ensemble of pads plus resonators. Figure 7 shows the experimentally obtained impedance curves vs. frequency of resonators with IDEs and pentagon spiral electrodes. The parameters of both resonators with IDEs and pentagon spiral electrodes are the same. The thickness of the Z-cut LiNbO_3_ thin film (*h_p_*) and electrode (*h_e_*) is set to 330 nm and 200 nm, respectively. The width of electrodes (*w*) is defined as 1 µm and the pitch (*p*) equals to 5 µm. The resonance frequency of XBAR with IDTs and pentagon spiral electrodes is 5.38 and 5.43 GHz, and $K_{eff}^{2}$ is 22.5% and 21.3%, respectively. The $K_{eff}^{2}$ of the resonator with pentagon spiral electrodes is almost equal to that of IDTs, which demonstrates that the electrode rotation angle has no impact on $K_{eff}^{2}$ of XBAR on Z-cut LiNbO_3_ thin film. Furthermore, it is also worth mentioning that the spurious modes in the impedance curve of XBAR with pentagon spiral electrodes are obviously suppressed, while the curve of resonator with IDEs structure is disturbed by spurious modes. In XBAR with IDE structure, the transverse waves are reflected at the edge of the electrode and form standing waves, thus causing spurious modes. However, in XBAR with pentagon spiral structure, due to the non-parallelism of electrode edges, the transverse waves have different reflection path at each point on the electrode edge, and the propagation path becomes longer, as shown in Figure 7b. When the transverse wave propagates through a long path, the energy of the standing wave is attenuated very low; therefore, the amplitude of spurious modes on the impedance curve are greatly reduced. It can also be noticed that the quality factor of XBAR with pentagon spiral electrodes is lower than that of XBAR with IDEs. The length of pentagon spiral electrodes is much larger than that of IDEs, which will increase the resistant of electrodes and eventually increase the electrical loss of electrodes.

Figure 8a shows the measured impedance curves of XBARs with different *p* of the top pentagon spiral electrodes. The resonance frequency is 5.433 GHz, 5.432 GHz and 5.230 GHz, respectively, when the *p* of electrodes is 5, 10 and 20 µm. The measured resonance frequency of the resonator with *p* = 20 µm is lower than that of *p* = 5 and 10 µm, which may due to the in-plane thickness inhomogeneity of LiNbO_3_ thin film, where the standing A_1_ modes are thickness-dependent when $h_{p}\ll p$. As *p* increases from 5 µm to 20 µm, the measured $K_{eff}^{2}$ decreases from 21.3% to 15.8% gradually as shown in Figure 8b. To better understand the variation of the $K_{eff}^{2}$ with different *p*, we adopt Berlincour’s Formulations (7) and (8) for electromechanical coupling calculation, which can be expressed as [27,28,29]
(7)$$K^{2}=\frac{U_{m}^{2}}{U_{e}U_{d}},$$
(8)$$U_{m}\propto \;\int^{\;}\left(E.d.T\right)\;dV,$$
where *K*^2^ is the electromechanical coupling, *U*_*m*_, *U*_*d*_, and *U*_*e*_ are the mutual energy, electrical energy, and elastic energy. *E*, *d*, *T* and *V* are the electric field, the piezoelectric coefficient, stress, and volume of the piezoelectric material, respectively. Equation (7) signifies the importance of the overlap between the applied electric field and the stress distribution of the resonant mode inside the piezoelectric material in the thickness direction. Figure 9 and Figure 10 show the stress distribution and total displacement distribution of A_1_ mode in the thickness direction of the LiNbO_3_ thin film, as the *p* = 5, 8, 10, 16 and 20 µm, respectively. The stress distribution and total displacement distribution declines with the increase of *p*, indicating the decrease in mutual energy and finally causing the diminution of $K_{eff}^{2}$.

## 4. Conclusions

In conclusion, we design and fabricate XBARs with pentagon spiral shape electrodes based on Z-cut LiNbO_3_ thin films. The proposed structure realized an A_1_ mode shear wave at 5.433 GHz along with $K_{eff}^{2}$ of 21.3%. The thickness of LiNbO_3_ thin film and the rotation angle of electrodes have no impact on the $K_{eff}^{2}$ of the proposed resonator. However, the $K_{eff}^{2}$ of XBAR is affected by the *p* of electrodes, decreasing from 21.3% to 15.8% with the increasement of *p* from 5 to 20 µm. Finally, a XBAR operates at 5.43 GHz with $K_{eff}^{2}$ of 21.5%, and little spurious mode is obtained when *w* = 1 µm and *p* = 5 µm. The almost no spurious mode resonator has great potential for 50 Ω impedance match within a miniature size applied in super high-frequency RF front-end filters.

## Figures and Tables

**Figure 1 micromachines-14-00552-f001:**
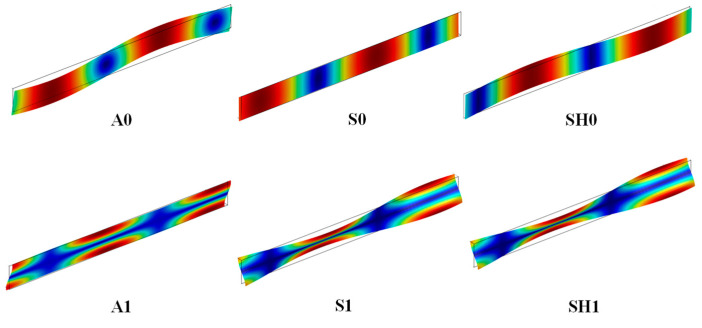
Schematic of the mode shapes of the first six plate modes (Lamb modes: A_0_, S_0_, A_1_, S_1_, and SH plate modes: SH_0_ and SH_1_) propagating in the Z-cut LiNbO_3_ membrane when $h_{\mathrm{LiNbO}3}/\lambda =0.1$.

**Figure 2 micromachines-14-00552-f002:**
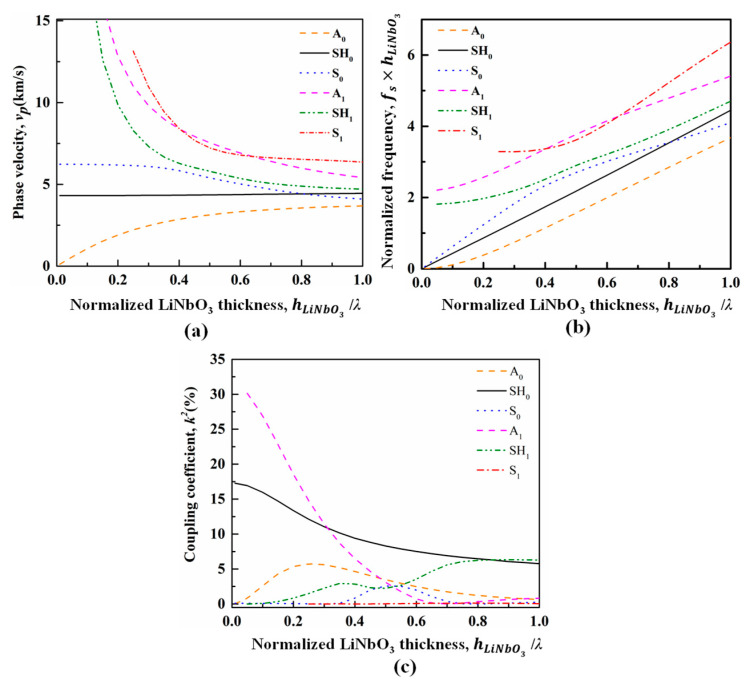
Calculated (**a**) dispersive phase velocities, (**b**) *f-β* dispersion curve, and (**c**) dispersive coupling coefficients of the first six plate modes in the Z-cut LiNbO_3_ membrane.

**Figure 3 micromachines-14-00552-f003:**
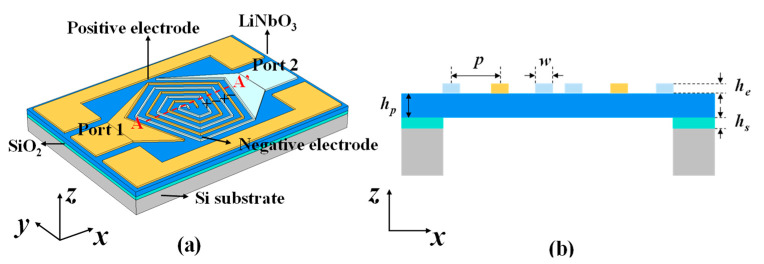
(**a**) Schematic drawing of XBAR with pentagon spiral shape electrode. (**b**) The A-A’ cross-section view of the resonator.

**Figure 4 micromachines-14-00552-f004:**
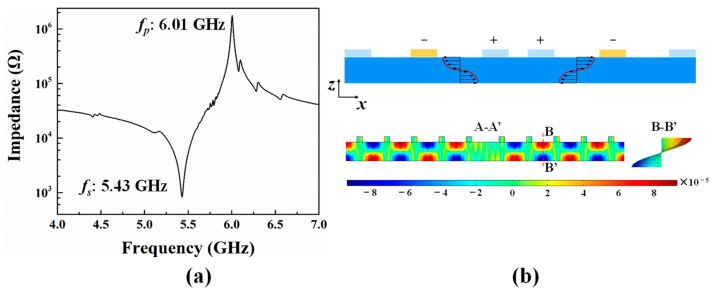
(**a**) The simulated impedance curve of resonator with pentagon spiral electrodes. (**b**) The simulated horizontal displacement mode shape at resonance frequency of A_1_ mode along the A-A’ cross section and the horizontal displacement deformation in the B-B’ cross section.

**Figure 5 micromachines-14-00552-f005:**
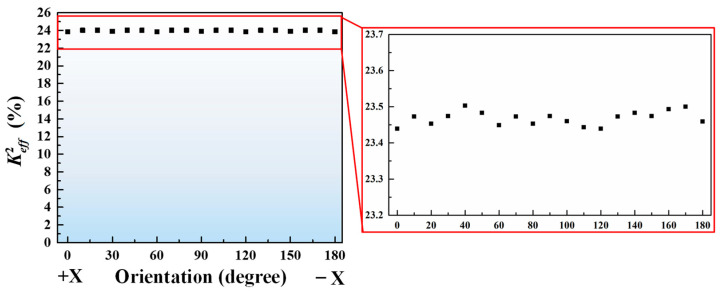
Calculated electromechanical coupling efficient of the A_1_ mode acoustic wave in a Z-cut LiNbO3 thin film vs. the rotation angle of electrodes.

**Figure 6 micromachines-14-00552-f006:**
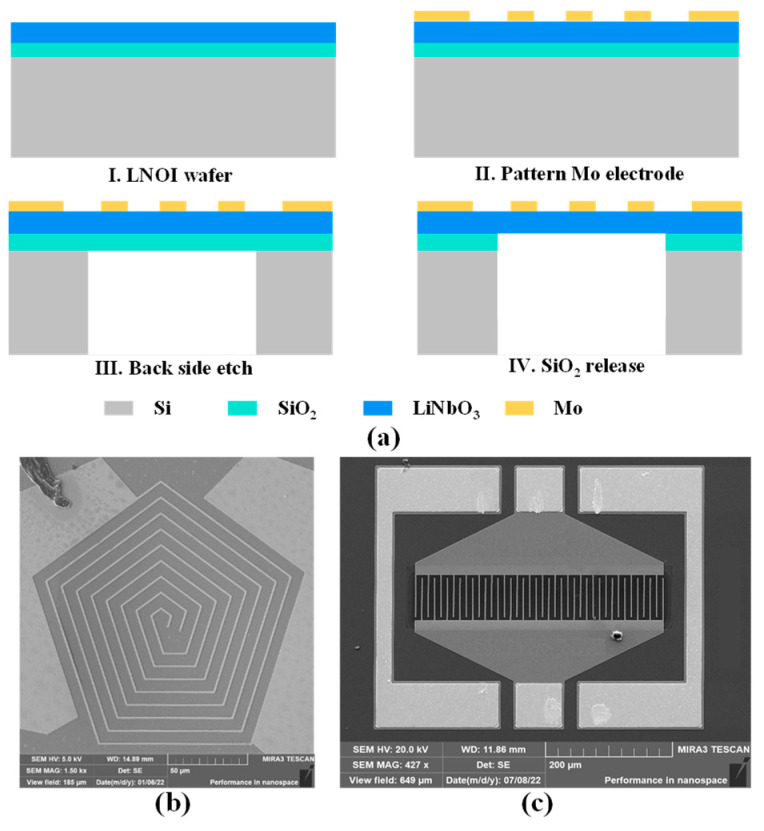
(**a**) Fabrication process flow for our resonator. I. Cross-section view of the LNOI wafer. II. Mo electrode pattern on the surface of LiNbO_3_. III. Backside Si deep reactive-ion etch. IV. Wet etch of buried SiO_2_ layer. (**b**) SEM image of the fabricated device with pentagon spiral electrodes. (**c**) SEM image of the fabricated device with IDEs.

**Figure 7 micromachines-14-00552-f007:**
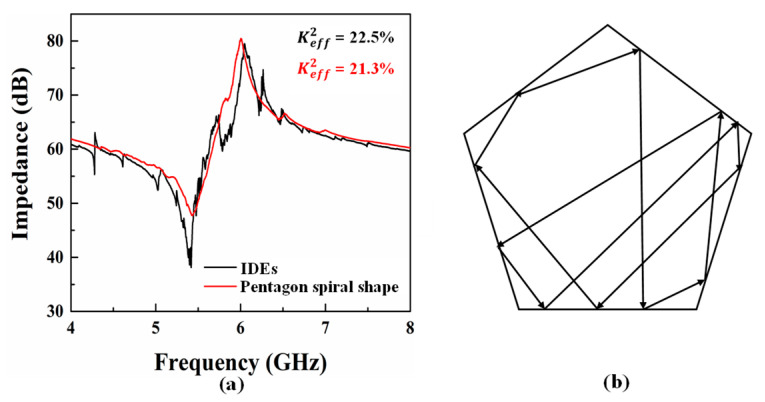
(**a**) The measured impedance curve of XBARs with IDTs structure (black) and pentagon spiral electrodes (red) vs. frequency, respectively. (**b**) Schematic diagram of acoustic wave propagation path in a pentagonal electrode.

**Figure 8 micromachines-14-00552-f008:**
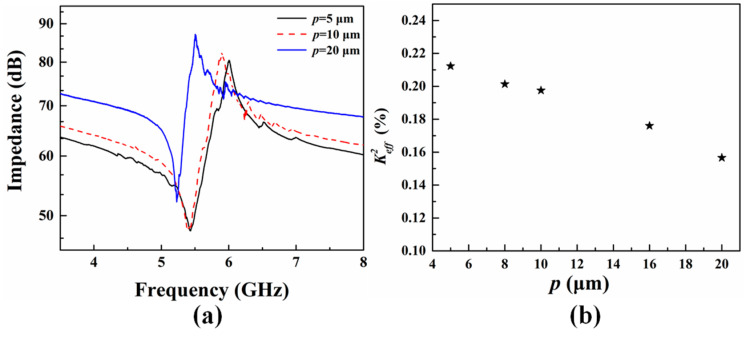
The measured impedance curve and $K_{eff}^{2}$ with different pitch (*p*). (**a**) The measured impedance curve of resonator with different pitch (*p*). (**b**) The measured $K_{eff}^{2}$ vs. different pitch (*p*).

**Figure 9 micromachines-14-00552-f009:**
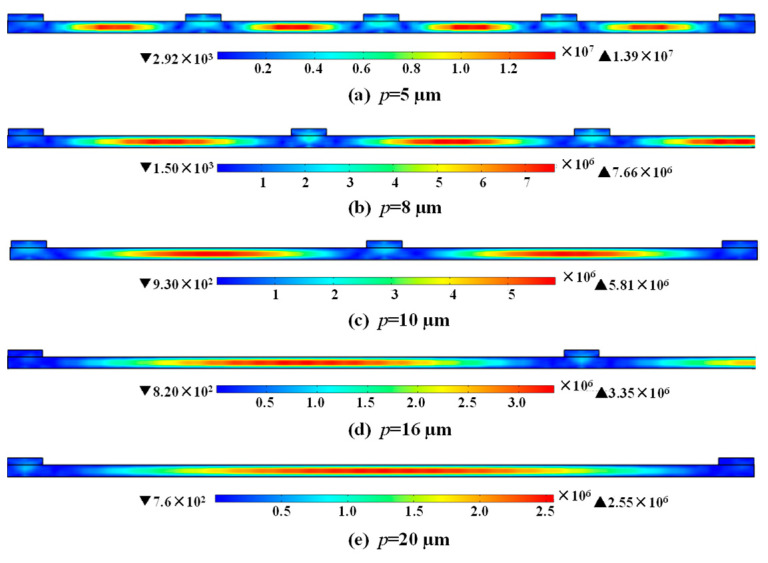
The simulated stress distribution at resonance as (**a**) *p* = 5 µm, (**b**) *p* = 8 µm, (**c**) *p* = 10 µm, (**d**) *p* = 16 µm and (**e**) *p* = 20 µm, while the width of electrode is maintained at 1 µm.

**Figure 10 micromachines-14-00552-f010:**
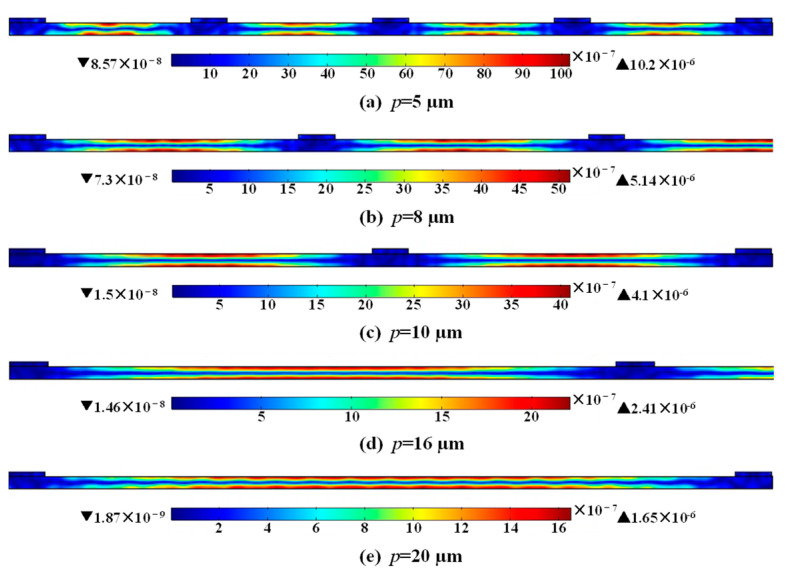
The simulated displacement distribution at resonance as (**a**) *p* = 5 µm, (**b**) *p* = 8 µm, (**c**) *p* = 10 µm, (**d**) *p* = 16 µm and (**e**) *p* = 20 µm, while the width of electrode is maintained at 1 µm.

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
