# Peer review of "Design and Analysis of Lithium–Niobate-Based Laterally Excited Bulk Acoustic Wave Resonator with Pentagon Spiral Electrodes"

_micromachines, 2023, doi:10.3390/mi14030552_

Round 1

Reviewer 1 Report

This paper describes a new type of BAW resonator excited on spiral electrodes. To the knowledge of the  reviewer it is a completely new approach and it should be published. Some general remarks are given below.

The authors claim this approach is free of spurious responses. To the reviewer point of view, it looks that spurious modes are present but they are different for the sections of the transducer with different angles. This results into smaller spurious at more frequencies. The paper would be improved if the reasons of reduced spurious are discussed.

Another thing which is not discussed is the increased effect of electrode resistance which reduce the Q at resonance frequency. Maybe, this should be mentioned. This can be seen clearly on figure 6. The IDT resonator has a much lower impedance at resonance than the spiral. It is probably because the electrodes have a smaller resistance and because several electrodes are in parallel reducing the equivalent series resistance.

p3 l115 replace "it can conclude" by "it can be concluded"  

Except these remarks, this is a good and original paper. A small  

Reviewer 2 Report

The manuscript titled "Design and Analysis of Lithium-Niobate-based Laterally Excited Bulk Acoustic Wave Resonator with pentagon spiral electrodes" by Xie et al. has presented the subject matter in a much precise way to cater the needs of readers and researchers. The article can be recommended for its possible publication, after some revisions.

1. Referring to figure 1, it would be useful for the reader to explain the abbreviations of the investigated plate modes.

2. Many variables/parameters are not defined in the text before recall them in equations/figures(i.e.: f_p, v_p, e_24, e_15,C_44, ...). Please revise it.

3. In row 87 the authors state "creating electric fields along both the x-direction and y-direction"; please provide the right references in the figures and in the text to better identify the configuration.

4. Figure 3.b is not clear. For clarity, is should include cartesian axys, and illustration on the right, bottom, should be explained. In this frame, it would be useful to show the simulated displacement mode shape along the whole cross section A-A', instead of the only central part of cross section.  

5. With present y scale, figure 4 does not properly show the variation of calculated electromechanical coupling coefficient of the A1 mode as a funcion of orientation. Please change it. 

6. Through the equation 2, the authors show the relation between K and piezoelectric coefficient in order to estimate the dependency of Keff^2 from orientation of electrodes (and shown in Figure 4), while through eqs. 5 and 6 they recall Berlincour's formulation for explain the dependence of electromechanical coupling from p. For coherence, at least a graph for this calculation should be included.

7. In Figure 8 the authors show the simulated stress distribution at resonance by varying the pitch. Can the authors show the displacement mode shape at resonance frequency at least for both the minimum and maximum pitch?

Reviewer 3 Report

Manuscript ID: micromachines-2233407

Title: Design and Analysis of Lithium-Niobate-based Laterally Excited Bulk Acoustic Wave Resonator with pentagon spiral electrodes

The authors report a laterally excited BAW resonator with pentagon spiral electrodes. The results show that the electromechanical coupling coefficient is independent of the electrode rotating angle. They fabricate the proposed laterally excited BAW resonator and find a resonating frequency of 5.433 GHz and an electromechanical coupling coefficient of 23.6%. The results are interesting and worthy of investigation. Publication in Micromachines could be considered only if the authors carefully address the following comments.

1.       Laterally excited BAW resonator possesses lots of advantages compared to SAW resonator. However, FBAR is also a potential candidate for high-performance resonators. The authors only mention the FBAR in one sentence (“Although AlN-based FBAR can achieve high frequency, the small electromechanical coupling coefficient does not meet the needs of large bandwidth [4-6].”) without any further comparison between the FBAR technique and the XBAR technique up-to-date. I strongly suggest the authors to introduce more about the FBAR and make a detailed comparison between FBAR and XBAR.

2.       Besides, there are already lots of literatures reporting about the XBAR. Though the authors summarize the research progress of XBAR properly, the advance and motivation of this work is not clearly stated. I recommend the authors to re-summarize the research progress, point out the advance and motivation of their work directly and apparently.

3.       About the details of FEM simulation method. The authors provide three published literatures for references. However, I think that for readers, it is still necessary to provide a brief description about the FEM simulation method. The readers therefore could understand how is the FEM simulation conducted. They could find the provided three references for more details if they are interested in.

4.       Page 4, Line 125. The “formula (5)” should be “formula (4)” I guess.

5.       In Figure 6, the authors compare the results of XBARs with IDT and pentagon spiral electrodes. What are the detailed parameters of IDT? Please provide them in the manuscript and a schematic figure is also suggested for better understanding.

6.       For the results shown in Figures 7 and 8, the authors use the simulation results to discuss the relation between electromechanical coupling coefficient and electrode pitch. Could the authors provide simulated impedance curves of the XBAR with different pitches, for the discussion of relation between resonant frequency and electrode pitch.

7.       Last question. The authors use the pentagon spiral electrodes to fabricate XBAR. What are the advantages of using this electrode structure instead of traditional IDT, other than the absence of spurious modes as illustrated in Figure 6? More interestingly, if the circular electrodes are adopted in XBAR, would the performance of XBAR be better compared to the one using pentagon spiral electrodes? The authors are encouraged to give some discussion.

Round 2

Reviewer 2 Report

The authors clarified all the critical points and improved the quality of the presentation. Hence the article can be recommended for its possible publication in its current form.